# Research on the Algal Density Change Features of Water Bodies in Urban Parks and the Environmental Driving Factors

**Yichuan Zhang** [1,2,*] , **Wenke Qin** [1] **and Lifang Qiao** [1,2]

1   School of Horticulture and Landscape Architecture, Henan Institute of Science and Technology, Xinxiang 453003, China
2   Henan Province Engineering Center of Horticultural Plant Resource Utilization and Germplasm Enhancement, Xinxiang 453003, China
*   Correspondence: zhangyichuan@hist.edu.cn

**Abstract:** Water bodies in urban parks are important for scenic and recreational areas, yet algal bloom problems seriously affect public use; therefore, it is important to study the features of algal density (AD) changes and environmental driving factors (EDFs) for water body management. In this study, five scenic water bodies in urban parks of Xinxiang City are taken as the objects for studying the AD and nine environmental indicators from March to October 2021, in accordance with time-series monitoring. The features of AD change in different layers of the water bodies are analyzed, and the main environmental impact factors of AD changes are screened by Pearson correlation analysis and principal components analysis (PCA), with main EDFs further extracted according to multiple linear regression analysis (MLRA), and multiple regression equation established. According to the data, ADs at different depth layers increase at first and then decrease with time, reaching the peak in August. According to the PCA, three principal components (PCs) are extracted in the 0.5 m and 1.0 m water layer, the variance contribution of which is 87.8% and 87.3%, respectively. The variance contribution of four PCs extracted in the 1.5 m water layer is 81.7%. After MLRA, it is found that the main EDFs of algal density in the 0.5 m water layer are electrical conductivity (EC), dissolved oxygen (DO), and water temperature (WT), in the 1.0 m water layer are WT and DO, and in the 1.5 m water layer are WT, DO, total nitrogen (TN), and EC. Generally speaking, WT and DO are decisive factors affecting AD. The EDFs' leads to the AD changes in different water layers are analyzed, and it is proved that stratification occurs in scenic water bodies in urban parks. This study is expected to provide basic data and a theoretical basis for ecosystem system protection and water quality management of scenic water bodies in urban parks.

**Keywords:** algal density (AD); environmental driving factors (EDFs); principal component analysis (PCA); aquatic ecosystem; water quality management

## 1. Introduction

Algae are important primary producers in the water ecosystem, a key link in the aquatic food chain, and an important bait resource for zooplankton and economically aquatic animals, playing an essential regulatory role in maintaining the ecological balance of the water body [1]. Algae can absorb nutrient salts, including nitrogen (N), phosphorus (P), etc., in the water body to synthesize materials for their own growth, but the eutrophication of the water body can enable the algae to grow in a rapid way, and to a certain extent, its development will trigger a group effect, resulting in algal blooms in the form of green paint covering the water surface due to accumulation, which has become a nerve-wracking issue in the traditional water treatment process [2]. Algal blooms usually cause decreases in water transparency, dissolved oxygen (DO) content, and biodiversity, which destroys the balance of water ecosystems. Consequently, it also has an impact on the water supply quality for domestic use as well as industrial and agricultural production, thus threatening the ecological safety around water bodies and causing widespread international concern [3,4].

In the northern part of Henan Province of China, water bodies in urban areas are mostly distributed inside some large parks in the form of artificial lakes. Water bodies in urban parks are an important part of urban water ecological scenery, with various ecological service values, including water supply, flood control, tourism, climate regulation, environment cleaning, maintenance of ecology and biodiversity, and the provision of recreation sites [5]. They are important for scenic and recreational areas. On the other hand, such water bodies often have small water areas, poor hydrodynamic conditions, low self-purification capacity, and are vulnerable to pollution, etc., resulting in problems such as black and odorous water, eutrophication, and other problems [6,7]. Eutrophic water bodies generate organic substances with unpleasant odors and release toxic substances into the water, affecting the function of the water body [8]. The decrease in water transparency caused by algae has an impact on the visual appearance [9], and the odor emitted seriously affects the satisfaction of park use [10]. Harmful algae can produce toxins (cyanotoxins), which cause acute or chronic health effects in mammals (including humans) and other organisms [11]. Accordingly, how to carry out efficient treatment has become a pressing issue.

Algal reproduction in lakes mainly occurs in stagnant areas, suggesting that water retention time could significantly affect algal growth [12]. Nutrient supply is the main factor affecting algal reproduction during an algal bloom, and temperature change may trigger algal reproduction, while high wind speed can cause turbulent mixing and lead to a short algal bloom [13]. The close link between benthic algae and macrophyte recovery indicates that benthic algae, as an indicator for quantifying the lake recovery process, may be more useful than nutrient levels [14].

The occurrence of algae is impacted by several environmental indicators, which can be roughly divided into physical and chemical indicators [15]. The network of algae group in each lake and river is unique, and environmental indicators play a vital role in the occurrence of algae [16]. However, different algae respond differently to the environmental indicators. For instance, water temperature (WT) and turbidity (Tu) are the major driving factors that lead to the occurrence of water booms of green algae *Cladophora* [17]. In the case of appropriate WT, Tu is the main driving factor. Electrical conductivity (EC) is an important explanatory variable that impacts algae reproduction in small and medium-sized reservoirs in subtropical cities in South China [18]. In a study carried out in the Taizi River Basin of Liaoning Province, suspended solids (SS) are the main environmental factor affecting the genera and species level, as well as the structure of the community and species distribution of diatom [19]. pH, DO and microalgae density are also of great importance, ultimately affect the growth rate and quality of microalgae products [20]. In addition, the inflow of N and P leads to algal blooms, which can lead to eutrophication of the water body [21]. A study on a park in Gwangju, South Korea, showed that heavy rainfalls resulted in the transport of total nitrogen (TN) and total phosphorus (TP) [22]. There is a certain relationship between rainwater quality and urban surface type, and the stormwater runoff in the park has high concentrations of chemical oxygen demand (COD), TN, and TP [23]. Controlling terrestrial N inputs and sediment suspension may be crucial to inhibiting the transition from grass-based to algal lake ecosystems [24]. Determining the thresholds of TP, submerged vegetation cover, and zooplankton size can provide a basis for ecological restoration in small shallow lakes and ponds [25].

Principal components analysis (PCA) and multiple linear regression analysis (MLRA) are commonly used approaches to reveal the mechanism of changes in water body quality. In [26], Li et al. identified the key drivers of eutrophication in Taihu Lake through PCA. There are strong spatiotemporal changes in the correlation between algae and water quality indicators, suggesting that the limiting factors that dominate algal growth depend on the seasonality and location. Therefore, it is necessary to reduce the input of N and P for long-term eutrophication control. PCA and MLRA are used to evaluate the contribution of climate, urbanization, and nutrient load to changes in water chemical characteristics [27]. Multivariate statistical analysis is applied to the evaluation of water quality and the analysis

of pollution sources in urban lakes [28]. Akbar et al. used PCA and GIS to analyze risk factors affecting groundwater quality [29].

Most of the above research objects are natural lakes, while there are few studies on scenic water bodies in cities. Water bodies in urban parks have important ecological and ornamental functions, but algae reproduction directly affects the performance of the comprehensive benefits. Due to the lack of relevant data support, the management of park water bodies has great blindness.

The aim of the study was to determine the AD changes features and main EDFs. Scenic water bodies in cities belong to relatively closed waters, thus the phenomenon of water stratification often occurs due to the poor fluidity. The density, physical and chemical indicators of algae in different water layers may vary greatly, so do the driving factors that lead to the changes in AD. Revealing this difference is therefore important for the efficient management of the water bodies in cities. Based on time-series monitoring data analysis of AD change features and the main EDFs, it can provide a scientific basis for water body management in urban parks, which is conducive to managers adopting corresponding emergency treatments or long-term governance methods according to the situation to maintain normal functions.

## 2. Materials and Methods

Four main steps were involved in this study. First, the research area and sampling point were selected. Second, the AD and the environmental indicators, including physical and chemical indicators, were measured. Third, Pearson correlation analysis was used to determine the correlation between the AD and environmental indicators. Fourth, PCA and MLRA were used to determine the main EDFs that cause AD changes (Figure 1).

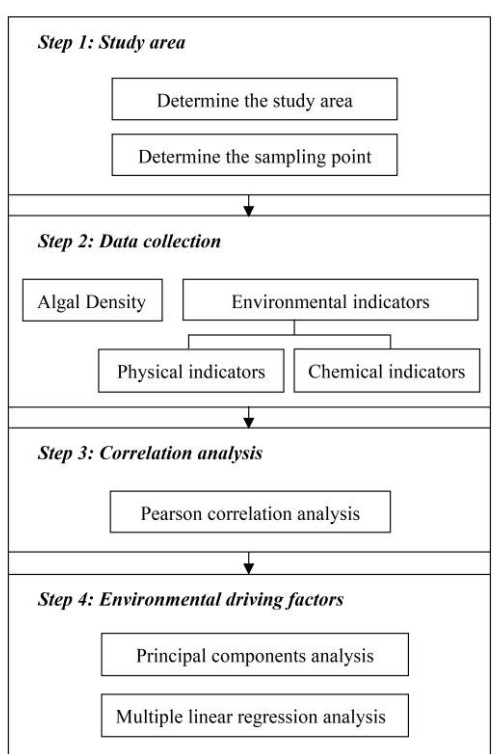

**Figure 1.** Workflow for the study.

### 2.1. Research Area Overview

Located in the northern part of Henan Province, China, Xinxiang City enjoys a warm temperate continental monsoon climate with four distinct seasons and has an average annual precipitation of about 573 mm [30]. The main water systems in Xinxiang include

the Wei River, the Communist Canal, the Mengjiangnv River, and the Zhaodingpai River, which play important roles in water diversion, irrigation, and drainage to the city. Water in the scenic and recreational areas in Xinxiang is mainly distributed in five parks with large areas: People's Park, Harmony Park, Muye Park, Xiangyang Park, and Dingguo Lake Park (Figure 2, Table 1).

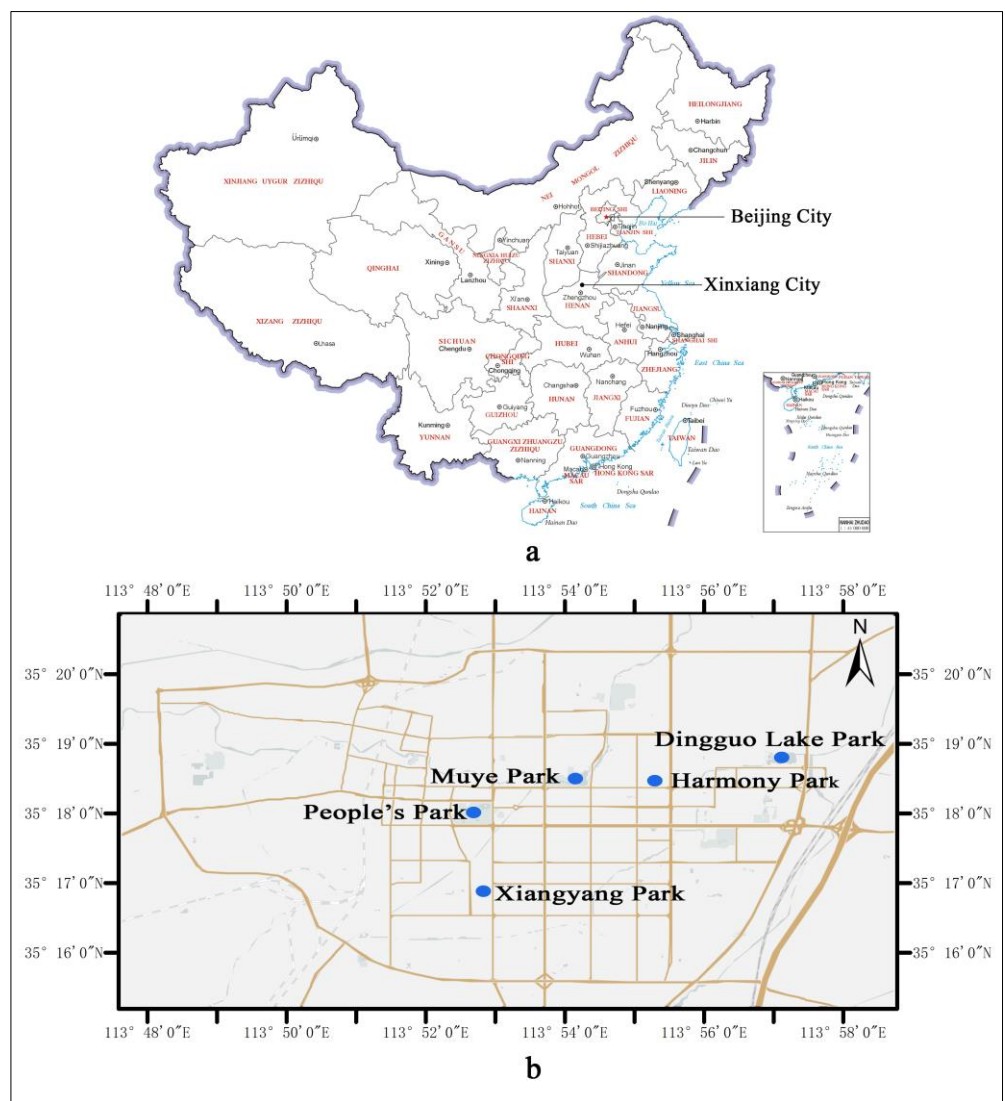

**Figure 2.** (**a**) Location of Xinxiang City. The standard map comes from the site http://bzdt.ch.mnr. gov.cn; (**b**) location of the five parks.

**Table 1.** Basic information about parks and lake.

| Name | Total Area (hm$^2$) | Lake Area (hm$^2$) | Water Depth (m) |
| --- | --- | --- | --- |
| People's Park | 35.1 | 3.43 | 2.5 |
| Harmony Park | 8.7 | 0.97 | 2.7 |
| Muye Park | 21.8 | 7.68 | 3.5 |
| Xiangyang Park | 3.0 | 0.12 | 2.1 |
| Dingguo Lake Park | 30.9 | 12.0 | 3.9 |

*2.2. Methods*

2.2.1. Algae and Environmental Indicators

The AD was measured using a YA-800L algae analyzer produced by Beijing Yi'an Science & Technology Company (Beijing, China). The environmental indicators were composed of five physical indicators—water temperature (WT), turbidity (Tu), suspended solids (SS), pH, and electrical conductivity (EC) and four chemical indicators—dissolved oxygen (DO), total nitrogen (TN), total phosphorus (TP), and chemical oxygen demand (COD). The Tu and SS contents were determined using a GLKRUI G968 water quality tester. The WT, EC, pH, and DO were measured using a SANXIN SX8336 portable water quality analyzer. The TP, TN, and COD were measured using a GLKRUI GL-800UVQ multi-parameter water quality analyzer and a GL-16 multi-functional intelligent dissipation instrument. In particular, the TP was determined in accordance with the water quality determination of the TP–ammonium molybdate spectrophotometric method (GB11893-89), the TN with the water quality determination of TN–alkaline potassium persulfate digestion–UV spectrophotometric method (GB/T 11894-1989), and the COD with the water quality determination of the COD fast D = digestion–spectrophotometric method (HJ/T399-2007).

2.2.2. Data Collection

The sampling and monitoring were conducted around the 21st of each month from March to October 2021 for the water bodies of People's Park, Harmony Park, Muye Park, Xiangyang Park, and Dingguo Lake Park. There were 3 sampling points set up for each park, and 1000 mL water samples were taken from the water layers of 0.5 m, 1.0 m, and 1.5 m using handheld water quality samplers (Figure 3). The AD and the 5 other environmental indicators of WT, Tu, SS, pH, and EC were measured onsite. About 500 mL of each water sample was extracted and filled into brown water bottles, and then brought back to the laboratory. Another 4 environmental indicators—DO, TN, TP, and COD—were determined in time using relevant instruments (Figure 4).

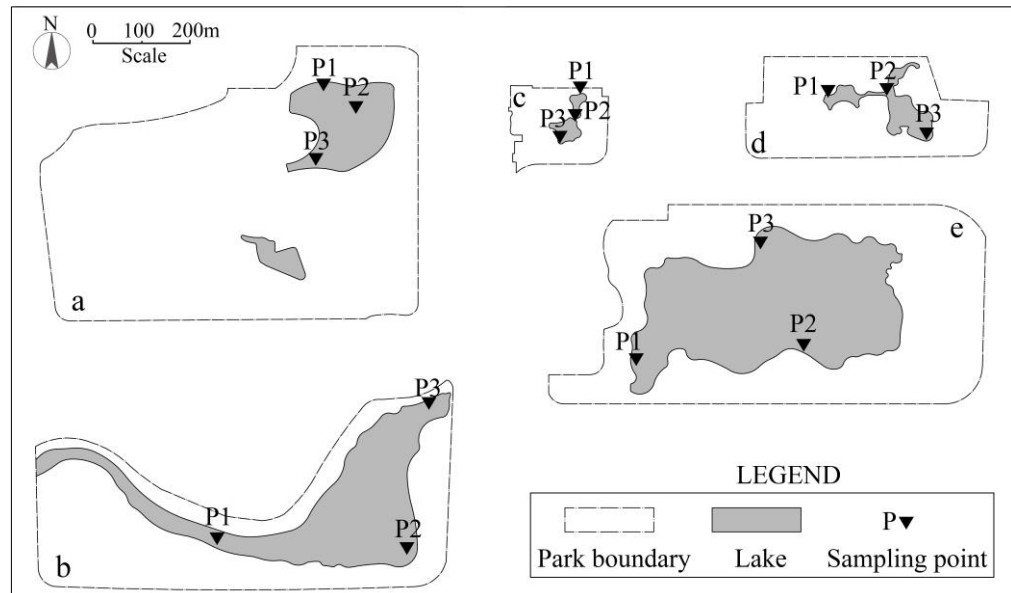

**Figure 3.** Location diagram of the sampling point. (**a**) People's Park; (**b**) Muye Park; (**c**) Xiangyang Park; (**d**) Harmony Park; (**e**) Dingguo Lake Park.

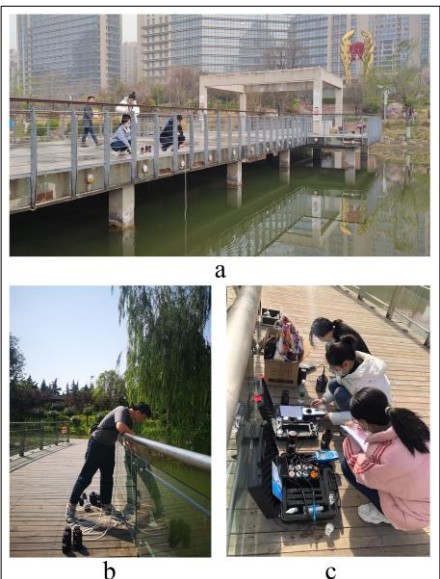

**Figure 4.** (**a**) Water sample collection in Harmony Park; (**b**) water sample collection in Xiangyang Park; (**c**) on-site determination of the WT, Tu, SS, pH, DO and EC (in Dingguo Lake Park).

### 2.2.3. Analysis Method

SPSS 21.0 software (Chicago, IL, USA) was used to conduct Pearson correlation analysis, PCA, and MLRA on the data. The Pearson correlation method was used for correlation analysis between the AD and the environmental indicators, as well as between the environmental indicators, to understand the interaction mechanism and screen out the main environmental impact factors. The main environmental impact factors were screened in the second round using the PCA method. Under normal circumstances, Kaiser–Meyer–Olkin (KMO) is suitable for PCA when it is greater than 0.5. KMO is an indicator for comparing simple relationships and partial relationships between variables. Usually, the higher the value, the stronger the correlation. Through PCA, the higher factor loadings in the PC of a high variance contribution rate were retained, while the high loading factors in the PC of a low variance contribution factor were deleted. MLRA was conducted with the AD as the dependent variable and the screened main environmental impact factors as the independent variables to acquire the main EDFs and establish quantitative expressions for the dynamic changes of AD under different water depth conditions.

## 3. Results

### 3.1. Inter-Monthly Change Features of AD

According to the time-series monitoring data, the inter-monthly changes of AD in the park-based water bodies were large (Figure 5). According to the figure, the ADs at different depth layers increased at first and then decreased with time, and all reached a peak in August. The AD changes in the 0.5 m layer were slower with time compared to those in the 1.0 m and 1.5 m layers. The ADs in the 0.5 m, 1.0 m, and 1.5 m layers from March to July illustrate a steady increasing tendency, increasing by about 3.2 times, 3 times, and 2.4 times, respectively. They peaked at a rapid increase from July to August and then began to plummet, all falling to about 4000 cells/mL in October.

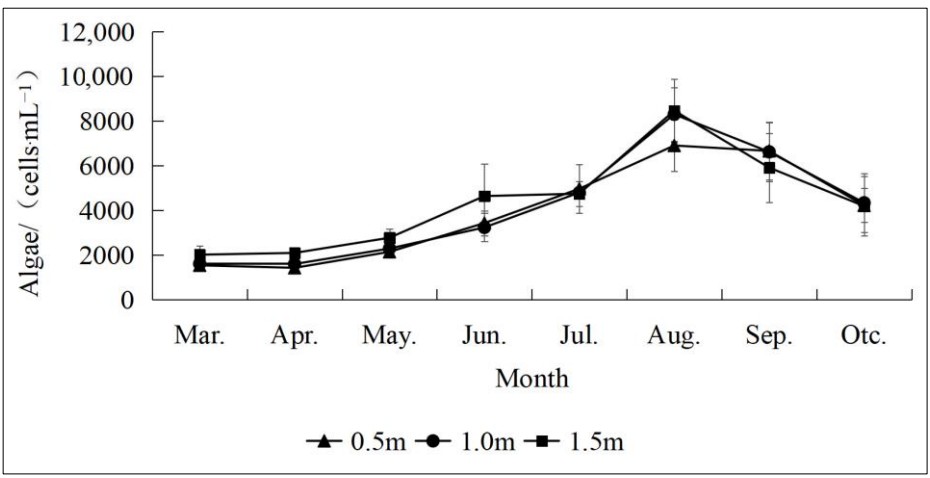

**Figure 5.** Inter-monthly change features of AD.

*3.2. Changes of Each Environment Indicator at Different Water Depths*

The change characteristics of each indicator at different water depths can be seen in Tables 2–4. The monthly fluctuation of the WT was not significant. The Tu increased with the increase in the water depth, while the Tu at different water depths tended to increase first and then decrease with the passing months. The SS increased with the increase in the water depth, while the SS at each water depth tended to increase until September and then decreased. The monthly fluctuation of the pH indicator at each water depth was not significant. The DO decreased with the increase in the water depths, while the DO at each water depth tended to decrease until June and then increased. The TN tended to decrease first and then increased with the increase in the water depth, while the TN at each water depth tended to decrease first and then increased with the passing months. The TP tended to decrease first and then increased with the increase in the water depth, while the TP at each water depth tended to decrease until June, increased until July and then decreased with the passing months. The COD generally tended to increase first and then decreased with the increase in the water depths. The fluctuation of the COD at each water depth with the passing months is relatively significant, reaching the relatively highest points in March, June, and August, and the lowest point in May.

**Table 2.** Values of each environment indicator at a 0.5 m water depth.

|  | WT (°C) | Tu (NTU) | SS (mg/L) | pH | EC (μS/cm) | DO (mg/L) | TN (mg/L) | TP (mg/L) | COD (mg/L) |
|---|---|---|---|---|---|---|---|---|---|
| Mar. | 14.89 ± 1.57 | 14.84 ± 11.93 | 15.60 ± 4.45 | 8.37 ± 0.32 | 142.1 ± 53.70 | 6.14 ± 1.07 | 0.162 ± 0.32 | 0.119 ± 0.41 | 41.07 ± 39.39 |
| Apr. | 19.02 ± 1.68 | 28.67 ± 32.19 | 19.80 ± 10.01 | 8.38 ± 0.24 | 121.1 ± 56.23 | 5.02 ± 0.38 | 0.081 ± 0.18 | 0.027 ± 0.05 | 12.93 ± 3.45 |
| May. | 27.57 ± 2.87 | 37.15 ± 39.26 | 21.80 ± 11.70 | 8.21 ± 0.27 | 124.1 ± 64.37 | 3.54 ± 0.34 | 0.108 ± 0.20 | 0.033 ± 0.06 | 9.00 ± 3.30 |
| Jun. | 30.65 ± 2.40 | 34.05 ± 30.54 | 21.40 ± 9.12 | 8.21 ± 0.41 | 128.7 ± 61.68 | 3.35 ± 0.29 | 0.177 ± 0.26 | 0.043 ± 0.03 | 19.07 ± 7.49 |
| Jul. | 31.74 ± 1.46 | 29.81 ± 40.97 | 20.00 ± 12.4 | 8.52 ± 0.84 | 42.35 ± 22.80 | 3.76 ± 0.86 | 0.244 ± 0.36 | 0.135 ± 0.24 | 11.07 ± 10.25 |
| Aug. | 27.93 ± 2.36 | 41.37 ± 45.55 | 23.67 ± 13.72 | 8.45 ± 0.76 | 66.71 ± 32.00 | 4.86 ± 1.06 | 0.818 ± 0.75 | 0.083 ± 0.13 | 20.07 ± 23.97 |
| Sep. | 23.51 ± 1.20 | 42.95 ± 40.85 | 24.07 ± 12.13 | 8.10 ± 0.34 | 56.13 ± 24.70 | 4.33 ± 0.45 | 1.14 ± 0.72 | 0.109 ± 0.16 | 17.73 ± 8.95 |
| Oct. | 20.83 ± 1.27 | 21.25 ± 16.26 | 17.67 ± 6.71 | 8.22 ± 0.36 | 93.85 ± 69.25 | 5.66 ± 0.48 | 1.48 ± 0.89 | 0.025 ± 0.02 | 13.83 ± 12.21 |

**Table 3.** Values of each environment indicator at a 1.0 m water depth.

|  | WT (°C) | Tu (NTU) | SS (mg/L) | pH | EC (μS/cm) | DO (mg/L) | TN (mg/L) | TP (mg/L) | COD (mg/L) |
|---|---|---|---|---|---|---|---|---|---|
| Mar. | 15.66 ± 2.22 | 15.03 ± 11.64 | 15.79 ± 3.51 | 8.36 ± 0.31 | 140.6 ± 56.78 | 6.07 ± 0.80 | 0.048 ± 0.09 | 0.039 ± 0.06 | 28.93 ± 23.07 |
| Apr. | 19.12 ± 1.79 | 35.16 ± 38.19 | 21.36 ± 11.86 | 8.34 ± 0.24 | 118.1 ± 57.03 | 5.09 ± 0.49 | 0.032 ± 0.07 | 0.029 ± 0.05 | 13.93 ± 3.43 |
| May. | 27.35 ± 2.82 | 41.51 ± 40.18 | 23.33 ± 12.26 | 8.21 ± 0.27 | 125.0 ± 62.69 | 3.61 ± 0.32 | 0.151 ± 0.26 | 0.015 ± 0.02 | 8.53 ± 3.68 |
| Jun. | 28.74 ± 7.66 | 53.89 ± 67.46 | 28.00 ± 19.87 | 8.15 ± 0.42 | 129.5 ± 66.15 | 3.37 ± 0.24 | 0.111 ± 0.22 | 0.033 ± 0.03 | 18.27 ± 7.90 |
| Jul. | 31.39 ± 1.64 | 33.49 ± 42.07 | 21.33 ± 13.06 | 8.32 ± 0.44 | 44.88 ± 57.12 | 3.57 ± 0.62 | 0.645 ± 0.52 | 0.136 ± 0.24 | 10.73 ± 11.22 |
| Aug. | 27.43 ± 2.85 | 43.63 ± 44.17 | 23.80 ± 13.17 | 8.47 ± 0.76 | 67.31 ± 35.62 | 4.83 ± 0.71 | 0.558 ± 0.75 | 0.081 ± 0.14 | 35.40 ± 57.43 |
| Sep. | 23.24 ± 1.88 | 44.49 ± 38.92 | 24.07 ± 11.69 | 8.09 ± 0.63 | 55.94 ± 25.76 | 4.29 ± 0.88 | 0.980 ± 0.52 | 0.109 ± 0.16 | 19.80 ± 8.93 |
| Oct. | 21.28 ± 1.53 | 19.78 ± 12.73 | 16.67 ± 3.97 | 8.26 ± 0.36 | 91.99 ± 62.27 | 5.61 ± 0.88 | 1.42 ± 0.90 | 0.061 ± 0.08 | 16.75 ± 10.22 |

**Table 4.** Values of each environment indicator at a 1.5 m water depth.

| | WT (°C) | Tu (NTU) | SS (mg/L) | pH | EC (μS/cm) | DO (mg/L) | TN (mg/L) | TP (mg/L) | COD (mg/L) |
|---|---|---|---|---|---|---|---|---|---|
| Mar. | 15.18 ± 1.55 | 23.12 ± 35.42 | 18.08 ± 10.56 | 8.30 ± 0.29 | 136.9 ± 60.73 | 5.85 ± 1.11 | 0.108 ± 0.19 | 0.067 ± 0.21 | 35.67 ± 27.71 |
| Apr. | 19.09 ± 1.83 | 59.83 ± 80.11 | 29.07 ± 23.58 | 8.32 ± 0.26 | 118.9 ± 57.98 | 4.91 ± 0.39 | 0.049 ± 0.11 | 0.037 ± 0.05 | 13.00 ± 2.11 |
| May. | 27.36 ± 2.91 | 51.04 ± 47.07 | 26.29 ± 14.10 | 8.17 ± 0.27 | 121.5 ± 63.25 | 3.67 ± 0.36 | 0.081 ± 0.19 | 0.009 ± 0.02 | 7.93 ± 3.00 |
| Jun. | 30.18 ± 1.08 | 83.33 ± 99.41 | 36.14 ± 29.97 | 8.07 ± 0.33 | 126.4 ± 66.53 | 3.50 ± 0.26 | 0.271 ± 0.34 | 0.054 ± 0.05 | 17.93 ± 5.57 |
| Jul. | 31.29 ± 1.38 | 46.59 ± 40.69 | 24.86 ± 12.39 | 8.26 ± 0.36 | 44.82 ± 51.06 | 3.49 ± 0.71 | 0.257 ± 0.56 | 0.152 ± 0.24 | 9.57 ± 10.41 |
| Aug. | 27.01 ± 2.93 | 55.53 ± 50.11 | 28.00 ± 15.26 | 8.29 ± 0.46 | 69.48 ± 36.21 | 4.55 ± 0.76 | 0.771 ± 0.76 | 0.092 ± 0.13 | 37.43 ± 54.74 |
| Sep. | 23.40 ± 1.75 | 46.67 ± 38.61 | 25.14 ± 11.40 | 8.04 ± 0.55 | 56.39 ± 26.23 | 4.17 ± 0.76 | 1.46 ± 0.74 | 0.115 ± 0.15 | 16.50 ± 13.73 |
| Oct. | 20.98 ± 1.53 | 28.72 ± 23.03 | 18.73 ± 7.29 | 8.19 ± 0.34 | 96.80 ± 66.68 | 5.36 ± 0.85 | 1.12 ± 0.82 | 0.028 ± 0.02 | 12.82 ± 10.42 |

### 3.3. Correlation Analysis between AD and Environmental Indicators

As seen in Table 5, the correlation between the AD and certain environmental indicators in the different water layers is high, among which the AD in the 0.5 m layer is highly significantly negatively correlated with the EC, and the AD in the 1.0 m layer is significantly negatively correlated with the EC. According to the correlation coefficients of the AD and environmental indicators, in the 0.5 m layer: EC > TN > SS > Tu > WT > TP > COD > DO > pH; in the 1.0 m layer: EC > COD > WT > Tu > SS > TP > TN > DO > pH; and in the 1.5 m layer: EC > COD > WT > DO > TP > SS > Tu > TN > pH. However, the environmental indicators highly related to the AD may have causal connections between themselves that have to be further refined based on the mechanism analysis by combining the correlation matrix between the environmental indicators.

**Table 5.** Correlation coefficients of AD and environmental indicators.

| | WT (°C) | Tu (NTU) | SS (mg/L) | pH | EC (μS/cm) | DO (mg/L) | TN (mg/L) | TP (mg/L) | COD (mg/L) |
|---|---|---|---|---|---|---|---|---|---|
| 0.5 m | 0.466 | 0.62 | 0.652 | −0.044 | −0.860 ** | −0.177 | 0.656 | 0.393 | −0.196 |
| 1.0 m | 0.438 | 0.347 | 0.301 | 0.102 | −0.811 * | −0.136 | 0.165 | −0.186 | 0.518 |
| 1.5 m | 0.504 | 0.242 | 0.248 | 0.072 | −0.695 | −0.29 | 0.104 | −0.282 | 0.512 |

** At the 0.01 level (two ends), the correlation is significant. * At the 0.05 level (two ends), the correlation is significant.

### 3.4. Correlation Analysis between Environmental Indicators

As can be seen in Table 6, the correlations between the following environmental indicators are high: the WT had a highly significant positive correlation with the Tu and SS, and the correlation coefficients are 0.527 and 0.526, respectively; for the DO, EC, and COD, the WT had highly significant negative correlations with them, and the correlation coefficients are −0.882, −0.467, and −0.406, respectively. This indicates that with an increase in the WT, the Tu and SS increased, while the DO and EC decreased, meaning that the increase in the WT had a certain inhibitory effect on the growth of the DO and EC in the water.

**Table 6.** Pearson correlation coefficient matrix between environmental indicators.

| | WT (°C) | Tu (NTU) | SS (mg/L) | pH | EC (μS/cm) | DO (mg/L) | TN (mg/L) | TP (mg/L) | COD (mg/L) |
|---|---|---|---|---|---|---|---|---|---|
| WT | 1 | | | | | | | | |
| Tu | 0.527 ** | 1 | | | | | | | |
| SS | 0.526 ** | 0.998 ** | 1 | | | | | | |
| pH | 0.272 | 0.1 | 0.092 | 1 | | | | | |
| EC | −0.467 * | −0.12 | −0.11 | −0.316 | 1 | | | | |
| DO | −0.882 ** | −0.627 ** | −0.627 ** | −0.233 | 0.261 | 1 | | | |
| TN | −0.029 | −0.118 | −0.13 | −0.111 | −0.570 ** | 0.206 | 1 | | |
| TP | 0.265 | −0.011 | −0.005 | 0.414 * | −0.743 ** | −0.128 | 0.255 | 1 | |
| COD | −0.406 * | −0.189 | −0.172 | −0.195 | 0.165 | 0.539 ** | −0.022 | 0.221 | 1 |

** At the 0.01 level (two ends), the correlation is significant. * At the 0.05 level (two ends), the correlation is significant.

The Tu and SS had a highly significant positive correlation, the Tu had a highly significant negative correlation with the DO, and the correlation coefficients are 0.998 and −0.627, respectively. This indicates that with the increase in the Tu, the SS increased while the DO decreased, meaning the increase in the Tu had a certain inhibitory effect on the DO in the water. The SS shows a highly significant negative correlation with the DO (−0.627), and the pH shows a significant positive correlation with the TP (0.414). The EC had a highly significant negative correlation with the TN and TP, with correlation coefficients of −0.570 and −0.743, respectively. The DO had a highly significant positive correlation with the COD (0.539).

### 3.5. PCA and Identification of Main EDFs

According to the KMO and Bartlett's test of sphericity for the screened environmental indicators, the KMO for different water layers were 0.575, 0.657, and 0.592, respectively, while the concomitant probabilities of different water layers from the Bartlett's test of sphericity were 0, which is less than the significance level of 0.05. These were considered suitable for the PCA. According to the PCA, the number of PCs was generally determined by the eigenvalues. To make the significance of each PC clearer, the maximum variance method was used to rotate the factors, and three PCs were extracted from the 0.5 m water layer, three PCs from the 1.0 m water layer, and four PCs from the 1.5 m water layer. According to the PCs, the cumulative variance contribution should be no less than 80% (Table 7). The factor loading matrix is shown in Table 8.

**Table 7.** Variance interpretation of variables.

| PC | 0.5 m | | | 1.0 m | | | 1.5 m | | |
|---|---|---|---|---|---|---|---|---|---|
| | Characteristic Root | VCR | CVCR | Characteristic Root | VCR | CVCR | Characteristic Root | VCR | CVCR |
| 1 | 2.2 | 43.994 | 43.994 | 2.265 | 45.292 | 45.292 | 2.286 | 28.581 | 28.581 |
| 2 | 1.13 | 22.598 | 66.592 | 1.08 | 21.604 | 66.896 | 1.822 | 22.774 | 51.355 |
| 3 | 1.059 | 21.171 | 87.763 | 1.021 | 20.425 | 87.321 | 1.388 | 17.347 | 68.702 |
| | | | | | | | 1.039 | 12.989 | 81.691 |

VCR: variance contribution rate; CVCR: cumulative variance contribution rate.

**Table 8.** Factor loading matrix.

| 0.5 m | | | | 1.0 m | | | | 1.5 m | | | | |
|---|---|---|---|---|---|---|---|---|---|---|---|---|
| EF | F1 | F2 | F3 | EF | F1 | F2 | F3 | EF | F1 | F2 | F3 | F4 |
| EC | −0.177 | −0.741 | 0.072 | EC | −0.167 | −0.153 | −0.7 | EC | −0.309 | −0.236 | −0.751 | 0.021 |
| TN | −0.357 | 0.732 | −0.157 | COD | 0.773 | −0.093 | 0.235 | COD | 0.001 | −0.076 | 0.031 | 0.944 |
| SS | 0.467 | 0.637 | 0.478 | WT | 0.049 | 0.901 | 0.174 | WT | 0.086 | 0.94 | 0.031 | 0.006 |
| Tu | 0.475 | 0.62 | 0.489 | Tu | 0.862 | 0.332 | 0.008 | DO | −0.214 | −0.904 | −0.072 | 0.059 |
| WT | 0.841 | 0.081 | −0.186 | SS | 0.859 | 0.348 | −0.003 | TP | 0.527 | 0.135 | 0.006 | 0.372 |
| TP | 0.066 | 0.075 | 0.87 | TP | 0.815 | −0.011 | 0.312 | SS | 0.961 | 0.136 | 0.09 | −0.037 |
| COD | −0.297 | −0.205 | 0.748 | TN | 0.104 | −0.062 | 0.843 | Tu | 0.963 | 0.13 | 0.084 | −0.043 |
| DO | −0.891 | −0.031 | −0.042 | DO | −0.197 | −0.871 | 0.071 | TN | −0.087 | −0.079 | 0.896 | 0.051 |

EF: environmental indicators.

According to Table 8, the first PC of the 0.5 m layer had the highest variance contribution of 43.994%. The DO and WT had higher loadings on the first PC, which mainly reflects the combined effect of both on the AD. The second PC had a smaller variance contribution of 22.598%. The loadings of the EC, TN, SS, and Tu were higher on this PC, which mainly reflects the combined effect of the EC, TN, SS, and Tu on the AD. The variance contribution of the third PC was 21.171%, and the high loading factors were the TP and COD, which mainly reflects the effects of the TP and COD on the AD. To sum up, the variance contribution of the third PC in the 0.5 m water layer was smaller compared to

the first and second PC, and the effects of the TP and COD on the AD are considered insignificant, so they were deleted. At this point, six indicators playing significant roles in the AD at the 0.5 m water layer were screened out, namely the DO, WT, EC, TN, SS, and Tu.

The first PC of the 1.0 m layer had the highest variance contribution of 45.292%. The loadings of the COD, Tu, SS, and TP on these PC were higher, which mainly reflects the combined effect of the COD, Tu, SS, and TP on the AD. The second PC had a smaller variance contribution rate of 21.604%, and the loadings of WT and DO on this PC were higher too, which also mainly reflects the combined effect of the WT and DO on the AD. The variance contribution of the third PC was 20.425%, and the high loading factors were the EC and TN, which mainly reflects the role of the EC and TN on the AD. To sum up, the variance contribution of the third PC in the 1.0 m water layer was smaller compared to the first and second PC, and the effects of the EC and TN on AD are considered insignificant, so they were deleted. Therefore, six indicators playing a significant role in the AD at the 1.0 m water layer were screened out, namely the COD, Tu, SS, TP, WT, and DO.

The first PC of the 1.5 m layer had the highest variance contribution of 28.581%. The loadings of the SS, Tu, and TP were higher on the first PC, which mainly reflects the combined effect of the SS, Tu, and TP on the AD. The second PC had a smaller variance contribution rate of 22.774%, and the WT and DO had higher loadings on this PC, which mainly reflects the combined effect of both on the AD. The variance contribution of the third PC was 17.347%, and the high loading factors are TN and EC, which mainly reflects the effect of the TN and EC on the AD. The variance contribution of the fourth PC was 12.989%, and the highest loading factor was COD, which mainly reflects the effect of the COD on the AD. To sum up, the fourth PC of the 1.5 m water layer had a smaller variance contribution rate compared to those of the first, second, and third PC, respectively, and the effect of the COD on the AD is considered insignificant, so it was deleted. Therefore, seven indicators playing a significant role on the AD in the 1.0 m water layer were screened out, namely the SS, Tu, TP, WT, DO, TN, and EC.

### 3.6. Establish Quantitative Relationship Equations

Multiple regression equations were established between the screened main environmental impact factors and the AD, with significance testing included, and the independent variables with probability values of $p > 0.05$ for significance testing were considered insignificant in relation to the dependent variable and deleted. As shown in Table 9, the TN, SS, and Tu were excluded from the 0.5 m layer; the COD, Tu, SS, and TP were excluded from the 1.0 m layer; and the SS, Tu, and TP were excluded from the 1.5 m layer. The results show that the main EDFs of the AD in the 0.5 m water layer were the EC, DO, and WT; the main EDFs of the AD in the 1.0 m water layer were the WT and DO; and the main EDFs of the AD in the 1.5 m water layer were the WT, DO, TN, and EC.

**Table 9.** Regression coefficients and significance test.

| | 0.5 m | | | 1.0 m | | | 1.5 m | |
|---|---|---|---|---|---|---|---|---|
| Regression Equation Factor | Coefficient | *p* Value | Regression Equation Factor | Coefficient | *p* Value | Regression Equation Factor | Coefficient | *p* Value |
| (Constant) | −4955.277 | 0 | (Constant) | −9038.279 | 0.036 | (Constant) | −17,006.221 | 0.011 |
| DO | 1209.403 | 0.001 | COD | −4514.945 | 0.267 | SS | 646.217 | 0.114 |
| WT | 268.912 | 0 | Tu | 7.091 | 0.904 | Tu | −193.614 | 0.114 |
| EC | −28.991 | 0 | SS | −25.291 | 0.898 | TP | −3873.883 | 0.115 |
| TN | 599.696 | 0.138 | TP | 613.855 | 0.889 | WT | 344.519 | 0 |
| SS | −18.705 | 0.928 | WT | 328.707 | 0 | DO | 1692.903 | 0.001 |
| Tu | −11.963 | 0.849 | DO | 1258.643 | 0.005 | TN | 1846.782 | 0 |
| | | | | | | EC | −23.687 | 0 |

The remaining independent variables were again established as multiple regression equations and tested for significance, and the final multiple regression equations were obtained as:

$$y\ (0.5\ \text{m}) = -7168.9 - 28.382\ \text{EC} + 1505.584\ \text{DO} + 282.911\ \text{WT}$$

$$y\ (1.0\ \text{m}) = -9884.43 + 328.757\ \text{WT} + 1325.007\ \text{DO}$$

$$y\ (1.5\ \text{m}) = -9943.12 + 336.71\ \text{WT} + 1668.186\ \text{DO} + 1912.728\ \text{TN} - 23.117\ \text{EC}$$

where y is the AD (cells/mL); EC is the electrical conductivity (μS/cm); DO is the dissolved oxygen (mg/L); WT is the water temperature (°C); and TN is the total nitrogen (mg/L).

## 4. Discussion

Reproduction, growth, and development of all aquatic species is impacted by WT [31]. WT is universally acknowledged to be one of the indicators that result in the decrease in the clarity of the lake, the primary cause is increasing WT promoting the growth of algae [32]. From the analysis of the sequence sampling results of the park water bodies, the ADs of the different water layers from March to July show a trend of increasing month by month, with a sudden increase and peak in July to August. Algae reproduce rapidly with a rise in temperature and are very prone to water blooms of blue-green algae. From August to October, the AD in the different water layers decreased rapidly with decreases in the WT. The average WT in the water bodies of the parks in Xinxiang City increased from 15.25 °C to 27.47 °C between March and August and then decreased to 21.03 °C in October. Since most algae are mesophilic [33], both low and high temperatures inhibit growth [34]. According to the research on Dianshan Lake, when the WT is below 18 °C, the abundance of blue-green algae remains unaffected by the physical and chemical indicators of water, regardless of the values of these parameters. However, when the WT is above 18 °C, the physical and chemical indicators of water play an important role in influencing the abundance of blue-green algae [35]. AD increases as WT increases within the range of 15–30 °C [36]. This WT range may be higher, up to 35 °C, in the tropics. Rising WT may also increase the degradation of organic matter and EC, leading to algal blooms [37]. Usually, the WT is higher in summer, resulting in a higher AD [38]. An elevated temperature has a significant effect on the life activity of algae because the increase in temperature promotes the enzymatic reaction of photosynthesis or the intensity of respiration, which directly affects the proliferation of algae [39]. The above phenomenon indicates that the WT was one of the main controlling factors for the increase in the AD, which is consistent with the results of the study concerning algal blooms in the National Wetland Park of Beijing Cuihu Lake conducted by Mao et al. [40]. The AD in the water bodies of the parks in Xinxiang City ranged from 2000–8000 cells/mL. The water bodies in the parks are turbid and no serious water blooms have been observed, which is consistent with the quantity level before the water bloom outbreak in the study of Liu et al. [41], and lower than the quantity level of the water bloom outbreak in the study of Zhou et al. [42].

The Pearson correlation analysis between the algae and the environmental indicators shows that the AD and the EC were significantly correlated in two water layers and correlated with a variety of environmental indicators. Shi et al. used the maximal information coefficient to quantitatively analyze the nonlinear relationship between the environmental indicators and algae in the outer waters of Dian Lake (in Yunnan Province), as well as Pearson linear correlation analysis to obtain more accurate correlation results between the algae and environmental indicators [43]. As AD interacts with multiple environmental indicators, it could not be explained by a single environmental factor. Water bodies in urban areas show the characteristics of both natural rivers and lakes, and the issue of algal blooms to them may exhibit multi-factor interactions [44]. Studies on Lake Hawassa in Ethiopia have shown that the algal dynamic change is heavily impacted by the EC, DO, pH, Tu, TP, and other parameters [45]. Therefore, it is necessary to study the combined effects of physical and chemical indicators on algal growth and competition simultaneously.

Human disturbances can also affect the results of the correlation analysis [46]. The water for the scenic and recreational areas of the parks is supplemented by adding tap water

or reclaimed water in addition to natural rainfall recharge in order to maintain a certain water level and ornamental scenic qualities. Eutrophic substances in the reclaimed water can increase the risk of water bloom outbreaks in different seasons [47,48]. Furthermore, extreme rainfall during the summer flood season can interfere with the results by causing water to spill into the municipal drainage system after a rapid rise. In normal circumstances, the water quality is generally worse in the wet season than in the dry season, mainly due to a significant increase in the ammonia nitrogen (NH4 + -N), SS, and TP concentrations after rain [49].

Among the main driving environmental indicators of AD in park water bodies, the WT, DO, and EC were the primary factors in all three water layers. Liu et al. found that the WT and DO were the main EDFs in a study on the driving factors of the seasonal succession of cyanobacterial composition in Chenghai—a plateau lake [50]. The DO mainly reflects the pollution level of organic matter, which is related to the WT, air pressure and other factors, and it is an important indicator in water environment assessment that hypoxia may occur if the DO in water is lower than 3~4 mg/L [51]. For Xinxiang City, the interval of the DO of water bodies in the parks from March to October was 3.4~6.03 mg/L, which decreased gradually with the increase in the WT and the lowest value was close to the state of hypoxia. Although the overall change of the DO shows that it was not too large, the sensitivity was high, which was one of the main EDFs. The EC reflects the ion concentration in the water column, and its increase may result in an increase in soluble nutrients in the water body [52].

The TP and TN have been the most important EDFs of the algal dynamic change of water bodies in many studies [53,54], but not always, as the response of different algae to nutrients varies [55,56]. Among the main EDFs of the AD changes of the water bodies in the parks of Xinxiang City, the TN was only present in the 1.5 m water layer. This is because the main sources of pollution in natural lakes are agricultural and breeding pollution, industrial and domestic sewage discharges, etc., while most of the water bodies in the parks of Xinxiang City are built in the central part of the park and are less disturbed by the outside world, and the main source of pollution is the nutrients brought by rainwater runoff. The interval of the TP from March to October in the water bodies of Xinxiang City Park was 0.019~0.141 mg/L, and the TN interval was 0.055~1.357 mg/L, which belong to Class III and Class IV surface water, respectively, according to the *National Standard of the People's Republic of China—Environmental Quality Standards for Surface Water (GB3838-2002)* [57]. On the other hand, the TP content exceeds its Class C standard and fails to meet the needs of water for recreation areas according to the *National Standard of the People's Republic of China—Water Quality Standard for Scenery and Recreation Area (GB12941-91)* [58].

The study has its limitations. Firstly, five scenic water bodies in urban parks of Xinxiang City are regarded as an integral whole, and the EDFs affecting AD changes in different water layers of the water bodies are focused. However, this may vary from park to park. Further studies need to analyze the data separately for each of the five parks. Although the EDFs that lead to AD changes have been found, which reasons affect these driving factors require further analysis, such as the discharge of domestic sewage, the collection of surface runoff, the bait fed, the design-caused dead angle, and the pollution of groundwater. Future research is needed to further analyze and clarify these factors by investigating the surroundings of the parks and interviewing management agencies, so as to improve the management efficiency and sustainability of scenic water bodies in urban parks.

## 5. Conclusions

(1) The ADs of different layers of the water bodies in the parks increased at first and then decreased during the study, and all of them reached their peaks in August. It was found that the trend of the AD in the 0.5 m water layer was relatively flat compared to those of the 1.0 m and 1.5 m water layers.

(2)   According to the PCA, three PCs are extracted in the 0.5 m and 1.0 m water layer, the variance contribution of which is 87.8% and 87.3%, respectively. The variance contribution of four PCs extracted in the 1.5 m water layer is 81.7%. The main environmental impact factors of the AD in the parks of Xinxiang City were the SS, Tu, TP, WT, DO, TN, EC, and COD.

(3)   After the MLRA and significance test, it was found that the main EDFs of the AD in the 0.5 m water layer were EC, DO, and WT; those in the 1.0 m water layer were the WT and DO; and those in the 1.0 m water layer were the WT, DO, TN, and EC. In general, the WT and DO were the most important parameters for AD.

(4)   The EDFs leads to the AD changes in different water layers are analyzed, and it is proved that stratification occurs in scenic water bodies in urban parks. This study is expected to provide basic data and theoretical basis for ecosystem system protection and water quality management of scenic water bodies in urban parks.

**Author Contributions:** Conceptualization, Y.Z. and L.Q.; methodology, W.Q.; software, W.Q.; validation, W.Q.; formal analysis, W.Q.; investigation, W.Q.; resources, W.Q.; data curation, W.Q.; writing—original draft preparation, W.Q.; writing—review and editing, Y.Z.; visualization, W.Q.; supervision, W.Q.; project administration, Y.Z.; funding acquisition, Y.Z. All authors have read and agreed to the published version of the manuscript.

**Funding:** This research was funded by the following projects: Key Science and Technology Research and Development Program of Henan Province, China (212102310841), Key Science and Technology Research and Development Program of Henan Province, China (212102310843), and Key Science and Technology Research and Development Program of Henan Province, China (222102320221). The APC was funded by the School of Horticulture and Landscape Architecture at the Henan Institute of Science and Technology.

**Institutional Review Board Statement:** Not applicable.

**Informed Consent Statement:** Not applicable.

**Data Availability Statement:** Data supporting the study are available from the corresponding author upon reasonable request.

**Conflicts of Interest:** The authors declare no conflict of interest.

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
