# Peer review of "Research on the Algal Density Change Features of Water Bodies in Urban Parks and the Environmental Driving Factors"

_sustainability, doi:10.3390/su142215263_

Round 1
Reviewer 1 Report
These manuscript needs to following major modifications:
1- Authors did not state novelties and stages of their study at the end of introduction.
2- Figure has not scale bar, latitude and longitude.
3- Authors did not state the annual values of DO, EC, COD and other quality parameters in different depths.
4- Authors did not draw flowchart of research methodology. Also, they did not state research methodology obviously.
5- Authors should show RMSE of applied method for simulation of algae.
Author Response
Dear Editor:
We would like to thank Sustainability for giving us the opportunity to revise our manuscript.
We would like to thank the reviewers for their thoughtful review of our manuscript. We believe that the additional changes we have made in response to the reviewers comments have made this a significantly stronger manuscript. Below is our point-by-point response to the referee’s comments.
Thanks for all the help.
Best wishes,
Yichuan Zhang
Corresponding Author
Responses to Reviewer 1
These manuscript needs to following major modifications:
1-Authors did not state novelties and stages of their study at the end of introduction.
RE: We added innovative explanations at the end of the Introduction. We added the description and figure of the research stages at the beginning of 2. Materials and Methods.
For detailed modifications, see line 85-101.
2-Figure has not scale bar, latitude and longitude.
RE: We resumed Figure 2, which increased the latitude and longitude.
For detailed modifications, see line 103-104.
3-Authors did not state the annual values of DO, EC, COD and other quality
parameters in different depths.
RE: We add three tables to explain the values of these indicators at different depths.
For detailed modifications, see line 164-185.
4-Authors did not draw flowchart of research methodology. Also, they did not state research methodology obviously.
RE: We reorganized the research method part, adding 2.2 Methods
5-Authors should show RMSE of applied method for simulation of algae.
RE: We added RMSE in the Figure 3.

Reviewer 2 Report
Research on the Algal Density Change Features of Water Bodies in Urban Parks and the Environmental Driving Factors
This is very interesting manuscript focusing on the development of algae in the urban aquatic environments. These water bodies are very important since they have role in water supply, flood control, tourism, climate regulation, environment cleaning, maintenance of biodiversity, and provision of recreation sites. Various environmental conditions affect algal development in the water bodies and high nutrient concentrations or water temperature may lead to the occurrence of the algal bloom. The presented results indicated that in the summer period algal communities reach highest abundance. These are valuable results which can contribute to the assessment of the environmental condition and management of the investigated urban water bodies. Unfortunately, the authors did not emphasize the established differences in environmental factors and algal density between the investigated sites. I think that major changes must be made before its publishing.
Specific comments are listed below.
Page 2, line 54: please use some a more frequent word than scholars (e.g. you may use scientist or some similar word)
Page 2, line 62: please use the abbreviation for total phosphorus in the sentence were you firstly mentioned this nutrient
Page 2, line 67: “… may trigger algal recovery…” – recovery from what? There is no clear and comprehensible connection between the sentences (line 66 – line 89) Please rewrite this part of the text with a clearly highlighted research goal
Page 2, line 70/71: Li Yongping et al. – is it reference 18? If yes, the author name should be corrected
Page 3, line 96: please add some reference after the first sentence in the par 2.1.
Page 3, part 2.2: it is common to firstly use full name of physical and chemical parameters in the sentence and after that their abbreviation. Please change this through the text.
Page 3, line 124-125: please use the abbreviation for investigated parameters
Page 3, line 127-128: please use the abbreviation for investigated parameters
Page 4, line 144: please delete the word Analyses form the title of the paragraph (it is sufficient to put the title Results)
Page 4, line 132: which significant level you used for Pearson correlation (p < 0.05 or?) please add the explanation in the legend in the Table 1
Page 4, line 158: it is not necessary to represent the same data in the text and in the table (Table 1), thus paragraph 3.2 could be shorter
Page 5, paragraph 3.3: please use the abbreviation for the environmental variables and I suggest shorten the paragraph as was mentioned earlier
Page 6, line 198: what is KMO? It should be described in the paragraph Analysis method
Generally, local environmental condition has significant influence on algal abundance. Since you collected samples from the five different water bodies, I suggest that differences between the studied sites should be presented. Accordingly, changes in environmental parameters at each water body should be shown and compared with density of algae at the same site. In line with this, Discussion and Conclusion should be rewritten.
Reference
Page 10, line 408: a comma is missing after the surname of authors
Page 10, line 411: a comma is missing after the surname of authors
Page 10, line 415-416: please correct the journal name and abbreviation, and use capital letter after the question mark (“the impact” should be changed into the “The impact”)
Page 10, line 417: please add a comma after the surname of all authors
Page 11, line 422: some author names are missing, please add a comma after the surnames Leavitt and Selbie
Page 11, line 425: a comma is missing after the last name of authors
Please check and correct the reference list.
Author Response
Dear Editor:
We would like to thank Sustainability for giving us the opportunity to revise our manuscript.
We would like to thank the reviewers for their thoughtful review of our manuscript. We believe that the additional changes we have made in response to the reviewers comments have made this a significantly stronger manuscript. Below is our point-by-point response to the referee’s comments.
Thanks for all the help.
Best wishes,
Yichuan Zhang
Corresponding Author
Responses to Reviewer 2
Page 2, line 54: please use some a more frequent word than scholars (e.g. you may use scientist or some similar word)
RE: We replaced this word.
Page 2, line 62: please use the abbreviation for total phosphorus in the sentence were you firstly mentioned this nutrient
RE: We have corrected the abbreviation.
Page 2, line 67: “… may trigger algal recovery…” – recovery from what? There is no clear and comprehensible connection between the sentences (line 66 – line 89) Please rewrite this part of the text with a clearly highlighted research goal
RE: “algal recovery” should be algal reproduction. Line 66-89 is mainly about the rules of algae and its influencing factors, we have compiled related research documents in this part.
Page 2, line 70/71: Li Yongping et al. – is it reference 18? If yes, the author name should be corrected
RE: We correct the way of expression.
Page 3, line 96: please add some reference after the first sentence in the par 2.1.
RE: We added 1 reference.
Page 3, part 2.2: it is common to firstly use full name of physical and chemical parameters in the sentence and after that their abbreviation. Please change this through the text.
RE: We have corrected all the abbreviations in the entire article.
Page 3, line 124-125: please use the abbreviation for investigated parameters
RE: We have corrected all the abbreviations in the entire article.
Page 3, line 127-128: please use the abbreviation for investigated parameters
RE: We have corrected all the abbreviations in the entire article.
Page 4, line 144: please delete the word Analyses form the title of the paragraph (it is sufficient to put the title Results)
RE: We deleted this word.
Page 4, line 132: which significant level you used for Pearson correlation (p < 0.05 or?) please add the explanation in the legend in the Table 1
RE: We added the explanation in the legend in the Table 1.
Page 4, line 158: it is not necessary to represent the same data in the text and in the table (Table 1), thus paragraph 3.2 could be shorter
RE: Thank you very much for your suggestion, but because the form is just some numbers, we added some explanation. This part has not been deleted to the content.
Page 5, paragraph 3.3: please use the abbreviation for the environmental variables and I suggest shorten the paragraph as was mentioned earlier
RE: We have corrected all the abbreviations in the entire article. Thank you very much for your suggestion, but because the form is just some numbers, we added some explanation. This part has not been deleted to the content.
Page 6, line 198: what is KMO? It should be described in the paragraph Analysis method
RE: KMO is a test parameter in PCA. We have added the interpretation of the KMO parameter as a explanation before the analysis.
Generally, local environmental condition has significant influence on algal abundance. Since you collected samples from the five different water bodies, I suggest that differences between the studied sites should be presented. Accordingly, changes in environmental parameters at each water body should be shown and compared with density of algae at the same site. In line with this, Discussion and Conclusion should be rewritten.
RE: Thank you for your suggestion. This is what we consider: this article is mainly to study the relationship between the algae density and environmental factors of the park's water body on the whole. From an external environment, the difference between these water bodies is not too large. We will analyze the differences between the water and bodies located in different parks in another paper in another paper, and try to analyze the cause.
Reference
RE: We carefully checked the problems existing in the reference, and made amendments according to the format of the journal.
Page 10, line 408: a comma is missing after the surname of authors
Page 10, line 411: a comma is missing after the surname of authors
Page 10, line 415-416: please correct the journal name and abbreviation, and use capital letter after the question mark (“the impact” should be changed into the “The impact”)
Page 10, line 417: please add a comma after the surname of all authors
Page 11, line 422: some author names are missing, please add a comma after the surnames Leavitt and Selbie
Page 11, line 425: a comma is missing after the last name of authors
Please check and correct the reference list.

Reviewer 3 Report
Manuscript entitled: Research on the Algal Density Change Features of Water Bodies in Urban Parks and the Environmental Driving Factors by Yichuan Zhang, Wenke Qin, Lifang Qiao give us some new information about density of algae in some Chinese reservoirs. However, the paper is still unclear, although it has been reviewed earlier.
1. The authors should compare the obtained results with data from other regions of the world, not only from Asia. 2. In line 115 Authors need to add some informations about morphometric characteristics of the reservoirs (i.g. area, deep…) 3. Some minor comments, I have marked in the text of the paper. After correcting, the manuscript can be published.
Author Response
Dear Editor:
We would like to thank Sustainability for giving us the opportunity to revise our manuscript.
We would like to thank the reviewers for their thoughtful review of our manuscript. We believe that the additional changes we have made in response to the reviewers comments have made this a significantly stronger manuscript. Below is our point-by-point response to the referee’s comments.
Thanks for all the help.
Best wishes,
Yichuan Zhang
Corresponding Author
Responses to Reviewer 3
1. The authors should compare the obtained results with data from other regions of the world, not only from Asia.
RE:We added some references and compared with the situation in other parts of the world.
For detailed modifications, see line 332-335, 343-354, 372-374, 399-401.
2. In line 115 Authors need to add some informations about morphometric characteristics of the reservoirs (i.g. area, deep…)
RE: We have added a table containing the park and water body.
For detailed modifications, see line 141.
3. Some minor comments, I have marked in the text of the paper. After correcting, the manuscript can be published.
RE: 1) We drew Figure 2 again;
2) We added the full name of abbreviation;
3) We moved the explanation of KMO to the method part;

Reviewer 4 Report
In the manuscript provided by Zhang and colleagues, the algal density change features of water bodies in urban parks and the environmental driving factors has been studied. In my opinion, although this article contains new aspects, the manuscript can be accepted with major amendments at Sustainability.
- English writing needs further polish.
- The abstract should be containing some quantitative results/findings. Also, the 'conclusion' section in the abstract part must be improved.
- The keywords provided by the authors are mainly derived from the main title. The authors should try to provide some different keywords. This would increase the visibility of the paper by search engines if accepted for publication by the journal.
- The novelty of this manuscript needs to be clearly stated in the introduction.
- The quality of the discussion section must be improved.
- The "literature review" section of the manuscript is poor. It is necessary to compare the results of the present study with previous similar studies.
- Limitations of the study must be presented in the conclusion section.
- For numbers in text and tables < 1.00, use three digits beyond the decimal point; for numbers between 1.00 and 9.99 use two digits beyond the decimal point; for numbers between 10.0 and 99.9, use one digit beyond the decimal point; and for concentrations ≥ 100, use the nearest whole number.
Author Response
Dear Editor:
We would like to thank Sustainability for giving us the opportunity to revise our manuscript.
We would like to thank the reviewers for their thoughtful review of our manuscript. We believe that the additional changes we have made in response to the reviewers comments have made this a significantly stronger manuscript. Below is our point-by-point response to the referee’s comments.
Thanks for all the help.
Best wishes,
Yichuan Zhang
Corresponding Author
Responses to Reviewer 4
- English writing needs further polish.
RE:We have used the English editing service provided by MDPI when submission. We have modified English expressions again. If the manuscript still cannot meet the language requirements, we will use the English editing service of another company.
- The abstract should be containing some quantitative results/findings. Also, the 'conclusion' section in the abstract part must be improved.
RE:We modified the abstract section, added a quantitative description to the result part, and the conclusion part was added.
For detailed modifications, see line 15-28.
- The keywords provided by the authors are mainly derived from the main title. The authors should try to provide some different keywords. This would increase the visibility of the paper by search engines if accepted for publication by the journal.
RE:We re-refined the keywords.
For detailed modifications, see line 29-30.
- The novelty of this manuscript needs to be clearly stated in the introduction.
RE:In the last paragraph of the introduction, we added the expression of the innovation point of the manuscript.
For detailed modifications, see line 109-114.
- The quality of the discussion section must be improved.
RE:We added some references and compared with the situation in other parts of the world. The number of references increases to 56.
For detailed modifications, see line 332-335, 343-354, 372-374, 399-401.
- The "literature review" section of the manuscript is poor. It is necessary to compare the results of the present study with previous similar studies.
RE:There are indeed many problems in our original review. We reorganized the review section, adding a dozen references, and deleting the relevant literature with a low correlation. The literature review is in the following order: the laws of algae changes, influence factors, and the application of PCA and MLRA methods. The number of references increases to 56.
For detailed modifications, see line 59-108.
- Limitations of the study must be presented in the conclusion section.
RE:We have increased the analysis of research limitations and described the focus of future research.
RE:For detailed modifications, see line 415-425.
- For numbers in text and tables < 1.00, use three digits beyond the decimal point; for numbers between 1.00 and 9.99 use two digits beyond the decimal point; for numbers between 10.0 and 99.9, use one digit beyond the decimal point; and for concentrations ≥ 100, use the nearest whole number.
RE: We refer to the data format in the following documents. We modified Table 1-3, but other forms still retain the format of software output.
https://www.mdpi.com/2071-1050/14/21/14572
https://www.mdpi.com/2071-1050/14/21/13940

Reviewer 5 Report
Many thanks for the review invitation of manuscript.
I read with great interest your paper entitled “Research on the Algal Density Change Features of Water Bodies in Urban Parks and the Environmental Driving Factors”. Urban parks often have water features that serve as scenic and recreational areas, but algal bloom issues can severely restrict public access. As a result, it's critical to understand the characteristics of algal changes and the environmental forces that influence them in order to manage water features. The findings indicate that in the 0.5 m water layer, electrical conductivity (EC), dissolved oxygen (DO), and water temperature (WT) are the main environmental drivers of AD change; in the 1.0 m water layer, WT and DO are the main environmental drivers; and in the 1.5 m water layer, WT, DO, total nitrogen (TN), and EC are the main environmental drivers.
Congratulations to the author with well written manuscript – However, I have proposed a “major revision” with the following comments:
Line 99: PCA, MLRA should define acronyms before using them. Please double-check all acronyms and definitions before using them.
Format the figure citations should follow the MDPI format, e.g. Figure instead of Fig.
The images' resolution should be increased, especially since the Legend are difficult to read.
Field images from the study area are required
Please provide a map of the sampling location
Conclusion is very shallow. It need to be improved.
Author Response
Dear Editor:
We would like to thank Sustainability for giving us the opportunity to revise our manuscript.
We would like to thank the reviewers for their thoughtful review of our manuscript. We believe that the additional changes we have made in response to the reviewers comments have made this a significantly stronger manuscript. Below is our point-by-point response to the referee’s comments.
Thanks for all the help.
Best wishes,
Yichuan Zhang
Corresponding Author
1. Line 99: PCA, MLRA should define acronyms before using them. Please double-check all acronyms and definitions before using them.
RE: We checked the full spells and abbreviations of proper nouns.
2. Format the figure citations should follow the MDPI format, e.g. Figure instead of Fig.
RE: We modified the format of the figure number.
3. The images' resolution should be increased, especially since the Legend are difficult to read.
RE: We redraw the Figure 2.
4. Field images from the study area are required
RE: We added field images (Figure 4). The position of the sampling point is marked.
5. Please provide a map of the sampling location
RE: We added Figure 3.
6. Conclusion is very shallow. It need to be improved.
RE: We modified the conclusion part.

Round 2
Reviewer 1 Report
This manuscript can be published by journal.
Author Response
Dear reviewer:
We use the english editing service provided by MDPI company to optimize the language expression of the paper.
Best wishes, Yichuan Zhang

Reviewer 2 Report
I think that authors made significant effort to improve the manuscript. Unfortunately, the authors stated that “… the difference between these water bodies is not too large” but it is not visible in the presented results. For example, in table 1 water temperature for March is 14.89 °C. Is it an average value from all investigated sites or maximal value measured? If it is an average value, I think that standard deviation must be showed or authors could present range values from xx to xx °C. Additionally, better description of the studied sites must be given (e.g. presence or not of macrophytes vegetation, the way of water uses, etc) to confirm that the water bodies are really very similar.
I think that it is better to show the influence of environmental factors on the number of algae in each lake separately, and then find out which parameters most significantly affect the number of algae in urban lakes.
The aim of the study should be clearly stated (e.g. The aim of the study was to find out the most important environmental parameters…. and then explain novelty).
In Conclusion, point 2 is not necessary, and in the point 3 it is obvious that WT and DO play most important parameters for algal density.
Author Response
I think that authors made significant effort to improve the manuscript.
(1) Unfortunately, the authors stated that “… the difference between these water bodies is not too large” but it is not visible in the presented results. For example, in table 1 water temperature for March is 14.89 °C. Is it an average value from all investigated sites or maximal value measured? If it is an average value, I think that standard deviation must be showed or authors could present range values from xx to xx °C. Additionally, better description of the studied sites must be given (e.g. presence or not of macrophytes vegetation, the way of water uses, etc) to confirm that the water bodies are really very similar.
RE: In tables 1, 2 and 3, we add the standard deviation of each environmental indicator.
(2) I think that it is better to show the influence of environmental factors on the number of algae in each lake separately, and then find out which parameters most significantly affect the number of algae in urban lakes.
RE: Thank you very much for your excellent suggestions, we will conduct more in-depth analysis in the follow-up study. But at present, the main focus of this paper is on the environmental driving factors of algal density changes in urban park water under different water depth conditions. Therefore, the driving factors of individual parks were not extracted in this study.
(3) The aim of the study should be clearly stated (e.g. The aim of the study was to find out the most important environmental parameters…. and then explain novelty).
RE: We add a statement in the last paragraph of the introduction that emphasizes the aim of the study.
(4) In Conclusion, point 2 is not necessary, and in the point 3 it is obvious that WT and DO play most important parameters for algal density.
RE: We removed the second point and added a summary statement to the third point.
(5) Language editing
RE: We use the English polishing service provided by MDPI company to optimize the language expression of the paper.

Reviewer 4 Report
The quality of the manuscript has been improved greatly so that it is adequate to publish in journal.
Author Response
Dear Reviewer:
Thanks for all the help.
Best wishes,
Yichuan Zhang
Reviewer 5 Report
Accepted in current form